# Characterization of the *MG828507* lncRNA Located Upstream of the *FLT1* Gene as an Etiology for Pre-Eclampsia

**DOI:** 10.3390/jcm11154603

**Published:** 2022-08-07

**Authors:** Hikari Yoshizawa, Haruki Nishizawa, Hidehito Inagaki, Keisuke Hitachi, Akiko Ohwaki, Yoshiko Sakabe, Mayuko Ito, Kunihiro Tsuchida, Takao Sekiya, Takuma Fujii, Hiroki Kurahashi

**Affiliations:** 1Department of Obstetrics and Gynecology, School of Medicine, Fujita Health University, Toyoake 470-1192, Japan; 2Division of Molecular Genetics, Institute for Comprehensive Medical Science, Fujita Health University, Toyoake 470-1192, Japan; 3Division for Therapies against Intractable Diseases, Institute for Comprehensive Medical Science, Fujita Health University, Toyoake 470-1192, Japan

**Keywords:** lncRNA, FLT1, placenta, pre-eclampsia

## Abstract

Background: *FLT1* is one of the significantly overexpressed genes found in a pre-eclamptic placenta and is involved with the etiology of this disease. Methods: We conducted genome-wide expression profiling by RNA-seq of placentas from women with pre-eclampsia and those with normotensive pregnancy. Results: We identified a lncRNA gene, *MG828507*, located ~80 kb upstream of the *FLT1* gene in a head-to-head orientation, which was overexpressed in the pre-eclamptic placenta. *MG828507* and *FLT1* are located within the same topologically associated domain in the genome. The *MG828507* mRNA level correlated with that of the *FLT1* in placentas from pre-eclamptic women as well as in samples from uncomplicated pregnancies. However, neither the overexpression nor knockdown of *MG828507* affected the expression of *FLT1*. Analysis of pre-eclampsia-linking genetic variants at this locus suggested that the placental genotype of one variant was associated with the expression of *MG828507*. The *MG828507* transcript level was not found to be associated with maternal blood pressure, but showed a relationship with birth and placental weights, suggesting that this lncRNA might be one of the pivotal placental factors in pre-eclampsia. Conclusion: Further characterization of the *MG828507* gene may elucidate the etiological roles of the *MG828507* and *FLT1* genes in pre-eclampsia in a genomic context.

## 1. Introduction

Pre-eclampsia is a syndrome defined by the onset of hypertension with proteinuria and is one of the most common obstetric problems, accounting for almost 15% of pregnancy-associated disorders [1]. It is not a simple complication of pregnancy, however, rather it is a syndrome of multiple organ failures involving the liver, kidneys, and lungs, in addition to coagulation and neural-system difficulties. Since cases of severe pre-eclampsia have a considerably poorer prognosis for both the mother and fetus than an uncomplicated pregnancy, it is potentially one of the most devastating pregnancy-associated disorders faced by gynecologists. There is now an emerging consensus that pre-eclampsia is a multifactorial disease, in which the pathogenetic processes underlying this disorder involve numerous factors such as oxidative stress, endothelial dysfunction, vasoconstriction, metabolic changes, thrombotic disorders, and inflammatory responses, although the precise mechanisms have continued to remain elusive [2,3].

It is generally accepted now that the placenta plays a primary role in the etiology of this disorder. A two-stage disease hypothesis has been proposed, in which an initiating reduction in placental perfusion by abnormal vascular remodeling leads to the maternal symptoms in individuals who have a genetic predisposition to this disease [4]. A considerable body of evidence indicates that placenta-derived anti-angiogenic factors, such as soluble fms-like tyrosine kinase-1 (sFlt-1) and soluble endoglin, are found at high concentrations in the maternal circulation in pre-eclampsia and significantly contribute to disease onset [5,6,7]. sFlt-1 is a truncated soluble form of this protein lacking the tyrosine kinase domain and acts as an anti-angiogenic factor, by neutralizing proangiogenic factors such as VEGF (vascular endothelial growth factor) or PlGF (placental growth factor), possibly leading to the abnormal placentation in the onset of pre-eclampsia. Assuming that pre-eclampsia is a polygenic disease, several lines of evidence suggest that genetic variants that increase the expression of the *FLT1* gene likely predispose the individual to pre-eclampsia. As the *FLT1* gene is located at chromosome 13q12.3, an elevated sFlt-1 and reduced placental growth factor (PlGF) are associated with trisomy 13 pregnancies, leading to a greater susceptibility to pre-eclampsia [8]. Nucleotide variants that are proximal to the *FLT1* gene in the placenta are also associated with this disease [9,10,11].

The fine-tuning of gene expression that contributes to the onset of polygenic diseases is often regulated by genetic variants, which act as an expression quantitative trait locus (eQTL). These variants might affect their target gene’s expression level via structural changes to a topologically associated domain (TAD) or via expression changes in lncRNAs (>200 nt) or miRNAs (<200 nt), which also regulate the expression of genes associated with disease [12,13]. Nearly 98% of human genome generates many species of non-coding RNAs, and it is now appreciated that many types of DNA regulatory elements, such as enhancers and promoters, produce lncRNAs to regulate the transcription of the relevant gene. Global gene-expression profile analysis is a powerful tool for the elucidation of the mechanistic pathway underlying the disease or identification of its diagnostic or prognostic markers. RNA-seq technology allows us to identify differentially expressed genes, including those genes that have not been well-characterized yet, such as lncRNAs [14,15,16]. In our current study, we conducted global-expression profiling using RNA-seq in women with uncomplicated pregnancies and in women with severe pre-eclampsia. Our comparative analyses focused on lncRNA species that might potentially regulate the expression of genes associated with pre-eclampsia.

## 2. Materials and Methods

### 2.1. Samples

All of the clinical samples analyzed in this study were collected at the Department of Obstetrics and Gynecology, Fujita Health University Hospital, Japan, from 2005 to 2014. Placental biopsy samples were obtained during Caesarean sections from both normotensive patients and women with severe pre-eclampsia (*n* = 39). Severe pre-eclampsia was defined by a blood pressure of greater than 160/110 mmHg and by proteinuria of more than 2 g over a 24 h collection period, according to the criteria used in Japan at that time, although these criteria have been revised, so the latest version is currently used [17,18]. Normotensive subjects (*n* = 38) were matched for both maternal and gestational age and for maternal body mass index at pre-pregnancy. Normotensive subjects underwent a Caesarean section due to a breech presentation or a previous Caesarean section. In addition, we collected preterm normotensive control samples from pregnancies with a premature rupture of the membrane that underwent a Caesarean section due to a breech presentation or with a previous Caesarean section without evidence of intrauterine infection. We calculated the birthweight coefficient by dividing the measured birthweight by the expected standard birthweight at the gestational week in the general Japanese population [19]. The clinical details of our study subjects are presented in Table 1.

To avoid any possible confounding effects of labor on gene expression, only placental samples that were obtained through Caesarean section from women who had not undergone labor were included in the analyses. A central area of chorionic tissue was then dissected, and the maternal deciduae and amnionic membranes were removed. We then dissected 1 cm sections of placental villi from the four different central areas between the basal and chorionic plates [20,21]. After vigorous washing of the maternal blood with saline, the tissues were immediately frozen in liquid nitrogen and stored until use. Informed consent was obtained from each patient, and this study was approved by the Ethical Review Board for Clinical Studies at Fujita Health University.

### 2.2. RNA-Seq

We performed RNA-seq analyses of placental tissues from severe pre-eclamptic (*n* = 6) and normotensive (*n* = 6) pregnancies. Total RNA was extracted from chorionic villous tissue samples using an RNeasy mini-kit (Qiagen, Valencia, CA, USA), in accordance with the instructions of the manufacturer. Briefly, approximately 1 μg of each total RNA sample was applied for library preparation using the NEBNext Ultra RNA Library Prep Kit for Illumina (#E7530; New England Biolabs, Inc., Ipswich, MA, USA), in accordance with the instructions of the manufacturer, except for the use of AMPure XP beads (A63880; Beckman Coulter, Indianapolis, IN, USA) in the clean-up steps. The NEBNext Poly(A) mRNA Magnetic Isolation Module upstream and the NEBNext Multiplex Oligos for Illumina downstream (#E7500, #E7335; New England Biolabs) were also used for poly(A) RNA purification and multiplex library production steps, respectively. The samples were sequenced using a high-throughput platform (HiSeq2500, Illumina) with a 100 bp single-end strategy.

We obtained about 40 million raw reads for each sample. These sequence data were quality-trimmed using FASTQX-Toolkit v0.0.13 (http://hannonlab.cshl.edu/fastx_toolkit/ accessed on 10 April 2022) with the command “-Q 33 -t 20 -l 30”. The reads were aligned to hg38 using Hisat2 ver. 2.0.5 with default parameters. Potential transcripts were assembled and read-counted using StringTie ver. 1.3.3 and TACO ver. 0.7.3 with gencode_v27_annotation.gtf file. The resulting counting data were used for the statistical analyses of differentially expressed genes by DESeq2 ver. 1.14.1 with the Wald test (cut-offs: baseMean > 10, false discovery rate (FDR; adjusted *p*-value, padj) < 0.05, and log2 fold change of >1 or <−1) [22].

### 2.3. Quantitative Real-Time RT-PCR

RNA recovered from the placental samples was subjected to quantitative real-time RT-PCR to quantify *MG828507* and *FLT1* gene expression. A Superscript III First-Strand Synthesis SuperMix for RT-PCR (Invitrogen, Grand Island, NY, USA) using random primers was employed to produce single-stranded cDNA from the total RNA. The 18S rRNA housekeeping gene 18S rRNA (Hs99999901_s1) was used to normalize the mRNA concentrations, because other genes commonly used as such a control are often regulated by estrogen [23]. RT-PCR reactions were performed in triplicate using a THUNDERBIRD Probe qPCR Mix (Toyobo, Osaka, Japan) in a final volume of 20 µL. The cycling conditions for PCR amplification were 1 min at 95 °C, followed by 40 cycles of 15 s at 95 °C, and 1 min at 60 °C. PCR primers and TaqMan probes for the *FLT1* gene were purchased commercially (Hs0105296_m1, Thermo Fisher Scientific, Tokyo, Japan), and those for the *MG828507* gene were custom-designed as follows: forward (5′-AAGTCAGCACACAGCTTGAAAGC-3′), reverse (5′-GTCTTGTGCTGTTTGACAAATGG-3′), and Taqman probe (5′-ACTACAGGCCTTTCTT-3′).

### 2.4. Overexpression of MG828507 in Cells

Almost the entire cDNA encoding *MG828507* was produced by RT-PCR and cloned into the pEGFP-N1 vector equipped with a CMV promoter. HTR-8/SVneo, a widely-used first trimester human trophoblast cell line, was purchased from Dr. Charles Graham at Queen’s University (Kingston, ON, Canada). The HTR-8 cells were seeded at 3.0 × 105 cells per 2 mL in 30 mm dishes. After 24 h, 3.2 µg of the vector was transfected using Lipofectamine 3000 (Thermo Fisher Scientific, Tokyo, Japan). HTR-8 cells that transiently overexpressing the *MG828507* gene were harvested at 48 h after transfection. Expression levels were assayed by qRT-PCR.

### 2.5. RNAi Knockdown of the MG828507 and FLT1 Genes 

HTR-8 cells were seeded at 3.0 × 105 cells per 2 mL in 30 mm dishes. After 24 h, siRNAs were transfected at a 40 nM concentration using Lipofectamine RNAiMax (Invitrogen, Grand Island, NY, USA). The Silencer™ Select siRNAs for the MG828507 gene were designed as follows; sense: 5′-UCUUUUUGUGAUGUAUGUGGC-3′, antisense: 5′-CACAUACAUCACAAAAAGAGG-3′. The siRNAs for the *FLT1* (Silencer™ Select Pre-Designed siRNA 4392420) and scrambled siRNA (Silencer™ Select Negative Control No. 1 siRNA 4390843) were purchased from Thermo Fisher. The HTR-8 cells were harvested at 96 h after transfection. Expression levels for the *MG828507* gene and the *FLT1* were assayed by qRT-PCR. 

### 2.6. Genomic Analysis 

Genomic DNA was extracted from the sampled placentas using a commercially available kit in accordance with the protocol of the manufacturer (Qiagen, Frankfurt, Germany). A total of 34 control samples from a normotensive pregnancy and 37 pre-eclampsia samples were used. Two single nucleotide variants (SNVs) (rs4769613, rs12050029) and one short tandem repeat (STR) variant (rs149427560), tag variants of each haplotype block located upstream of the *FLT1* gene, were genotyped. TaqMan primers and probes were purchased to genotype the rs4769613, rs12050029 SNVs, in accordance with the protocol of the manufacturer (C_32231378_10, C_1445411_10, Applied Biosystems, Foster City, CA, USA). For the STR variant, rs149427560, forward primers were labeled with FAM. The PCR products were analyzed by capillary electrophoresis (ABI3730 Genetic Analyzer; Applied Biosystems, Tokyo, Japan). Genotype deviations from a Hardy–Weinberg equilibrium (HWE) were first evaluated using the chi-squared test. Genotype and allele frequency differences between the pre-eclamptic and control groups were then evaluated using chi-squared analysis. All of these calculations were performed using SNPAlyze software (Dynacom, Chiba, Japan). Power calculations were performed using a genetic-power calculator. 

To analyze the topologically associated domain (TAD) surrounding the *FLT1* gene and the variants at that site, we used published Hi-C data for the trophoblast cell line (http://promoter.bx.psu.edu/hi-c/view.php accessed on 15 April 2022).

### 2.7. Statistical Analysis 

Intergroup comparisons were made using the Mann–Whitney U test or one way analysis of variance method, and *p* values of less than 0.05 were considered statistically significant. Correlations were evaluated using a Spearman’s test. In significant difference tests, *p* values were calculated with the z conversion of Fisher’s r. *p* values of less than 0.05 were again considered statistically significant.

## 3. Results

### 3.1. Identification of a lncRNA Gene Upregulated in Pre-Eclampsia

To further understand the molecular mechanisms underlying the symptoms of pre-eclampsia, we performed RNA-seq on pre-eclamptic placentas and normal uncomplicated pregnancies to compare their expression profiles. In comparisons of the data from six pre-eclamptic and six normotensive samples, we extracted 240 transcripts that were differentially expressed in the pre-eclamptic placenta (log2 fold-change >1 or <−1, FDR < 0.05). Since we focused on lncRNAs, we omitted protein-coding genes using annotation data in ref_gene_type and, thereby, obtained a gene set comprising 41 transcripts enriched for these nucleic acids. We manually characterized each transcript, examining the location and annotation information individually. We noted that one of the lncRNA genes in our panel, *MG828507*, was previously identified as being expressed in the placenta [24]. qRT-PCR confirmed that MG828507 was more abundant in the six pre-eclamptic placentas used for RNA-seq than in the controls.

*MG828507* contains two exons and encodes a 6489 bp lncRNA. It is located ~80 kb upstream of the *FLT1* gene in a head-to-head orientation (Figure 1). The 5′ upstream region and 3′ end of the first exon is conserved among mammals, but all other regions are conserved only among primates. *MG828507* is known to be abundant in the placenta, and RNA-seq data of various organs in the public database reveals that the expression is scarcely found in other tissues than placenta [24]. Since the *MG828507* and *FLT1* genes are located within the same TAD, we hypothesized that *MG828507* may affect the expression of *FLT1* and, thereby, have an impact on susceptibility to pre-eclampsia.

### 3.2. Comparison of MG828507 and FLT1 Expression in Pre-Eclamptic and Normotensive Placentas

To validate the differential expression observed in our RNA-seq data, we increased the sample numbers and performed qRT-PCR (Figure 2A). We confirmed the overexpression of *MG828507* in our pre-eclamptic placental samples (*n* = 39), compared to those from the uncomplicated normotensive control pregnancies (*n* = 38), and found that this difference was statistically significant (*p* < 0.01). We also examined the expression level of the *FLT1* gene and found it to be significantly high in our pre-eclamptic samples, as reported previously [5]. Notably, the expression of *MG828507* appeared to be linearly correlated with that of *FLT1* (Figure 2B).

### 3.3. Analysis of MG828507 and FLT1 Expression in the HTR-8 Trophoblast Cell Line

To test the possibility that the *MG828507* lncRNA may regulate the *FLT1* gene, we exogenously overexpressed it in the HTR-8 trophoblast cell line using a transfected vector but observed no impact on the *FLT1* expression level (Figure 3A). We then knocked down the *MG828507* gene in these same cells via siRNA transfection but again saw no changes in *FLT1* expression (Figure 3B). 

### 3.4. Analysis of MG828507 Expression in the Genetic Variants 

Next, we genotyped our placental samples for two SNPs (rs4769613 and rs12050029) and one STR (rs149427560) that are located around the *MG828507* gene and examined whether there was any association between the expression of this lncRNA and these variants. The genotype frequencies in our samples were comparable to those reported in East Asian populations, and the distribution satisfied the HWE. When we analyzed the association between *MG828507* gene expression and rs4769613 and rs12050029, via allele wise and genotype wise analysis, no association was observed (Figure 4). We then examined the STR repeat numbers. Among the four size variants (472, 474, 476, and 478) identified in our samples, the 474 and 476 alleles were observed more frequently than the others. In the genotype-wise analysis, the pre-eclamptic samples of the 476/476 homozygotes showed significantly higher *MG828507* expression (*p* < 0.05). The 476 allele was significantly more frequent than the 474 allele in the allele-wise analysis (Figure 4). 

### 3.5. Correlations between MG828507 Expression and Clinical Parameters

We finally assessed whether there were any correlations between *MG828507* expression and various clinical parameters that might reflect the severity of pre-eclampsia. With regard to disease onset, no significant difference in this expression was observed between the severe early onset group (earlier than 34 weeks of gestation, *n* = 22) and late onset group (34 weeks of gestation or later, *n* = 17) (Appendix A). The *MG828507* levels appeared to correlate, however, with both the systolic and diastolic blood pressure (Figure 5A,B). However, when these analyses were performed separately for each group, no correlation was observed within either the pre-eclamptic or normotensive groups. These findings suggest that the correlation between *MG828507* level and blood pressure simply reflects an association between this lncRNA and the presence or absence of pre-eclampsia. On the other hand, a negative correlation was observed between the *MG828507* level and the normalized birth or placental weights (Figure 5C,D). Even after stratification by group, this negative correlation was still evident. In addition, we analyzed the correlation between the *MG828507* level and other clinical parameters that indicate disease severity, including platelet count, serum transaminases, and creatinine levels. However, they did not correlate with the *MG828507* level (Appendix A). 

## 4. Discussion

There is growing evidence that the unbalanced expression of specific lncRNA is involved in the pathogenesis of pre-eclampsia, and several important lncRNA genes have been identified via RNA-seq strategy [15,25,26]. We have reported here that *MG828507* lncRNA is expressed in the placenta, in tandem with the *FLT1* gene, and is highly expressed in a pre-eclamptic placenta compared to that of an uncomplicated normotensive pregnancy. It has been amply documented that lncRNAs play diverse regulatory roles in gene expression [27]. A growing body of evidence now indicates that the lncRNAs act as regulators of placental development and differentiation [28]. Since the expression of *MG828507* lncRNA was found in our current analyses to be linearly associated with that of the *FLT1* gene, we postulated that *MG828507* lncRNA may have a regulatory role in the expression of *FLT1*.

The *MG828507* and *FLT1* genes are located within the same TAD. The TADs are megabase-scale genomic domains, in which certain DNA regions show a significantly higher interaction frequency with other DNA regions within the domain, compared with those outside of the domain [29,30]. TADs are generally separated by an insulating protein complex, including CTCF, and build a common framework for contact between regulatory elements and genes within the domain [31]. One possibility in relation to our current investigation is that *MG828507* is an enhancer of regulatory RNA for the *FLT1* gene. Hence, the promoter of the *FLT1* gene and *MG828507* may interact within the TAD, forming a transcriptional complex via a loop conformation. Although Genhancer analysis did not indicate any interaction between *MG828507* and *FLT1*, the public databases indicate a positive acetylation of H3K27 at the 5′ end of *MG828507* exon, suggesting a possible enhancer function of *MG828507* (data not shown).

As regulatory lncRNAs or enhancer RNAs generally regulate their downstream target genes in either a cis or trans manner, we conducted an overexpression experiment with *MG828507* as a gain-of-function analysis and also, in parallel, a knockdown using siRNA as a loss-of-function experiment [32]. The expression of the *FLT1* gene was unaffected in both instances, however, which did not support our hypothesis. One notable limitation of these experiments was the possibility that HTR-8 cells have already lost the fine regulation of *FLT1* expression that is predicted in normal trophoblasts, since the *FLT1* mRNA level is low and the *MG828507* level is quite high in these cells relative to a normal placenta (data not shown). Alternatively, synergistic action of other genetic modifiers might be required to observe the effects of *MG828507* on *FLT1*.

We analyzed three *FLT1* upstream genomic variants that were previously reported to be associated with pre-eclampsia [9]. In our previous study, two of the SNPs we again tested here did not show such an association, but the rs149427560 STR showed a weak association with pre-eclampsia [11]. Although it was not unreasonable to speculate that the association was possibly based on an eQTL effect by the STR upon *FLT1* expression, we did not observe any association between this genotype and the *FLT1* expression level in our previous investigation [11]. Notably, however, this STR was found in our current analyses to be associated with *MG828507* expression in the pre-eclamptic placentas. The 476/476 genotype, showing a higher *MG828507* level, was frequent in pre-eclampsia, while the 474/476 genotype, showing a lower *MG828507* expression in our present analysis, was found in our prior study to be less frequent in the pre-eclamptic population [11]. This suggests that this STR may serve as an eQTL for *MG828507* and, thereby, lead to a higher susceptibility to pre-eclampsia via the modification of *FLT1* expression. Further validation with an increased sample number will be necessary.

When we analyzed the correlation here between *MG828507* expression and the clinical parameters of our study subjects, no association was found with the maternal blood pressure, but a correlation was observed with both the birth and placental weights. Based on our observation that the expression of *MG828507* correlates that of *FLT1* anti-angiogenic factor, an augmented *MG828507* expression may induce sFlt-1 and, thus, suppress angiogenic factors such as VEGF or PlGF and cause abnormal placentation. It is generally accepted that the initial event in pre-eclampsia is a reduction in placental perfusion via abnormal vascular remodeling, which later leads to symptoms in the women who have a genetic predisposition to the disease [4]. It is not unreasonable, therefore, to speculate that an increase in *MG828507* or *FLT1* expression may be a fundamental placental event that leads to both fetal and placental symptoms, which might not correlate with maternal symptom severity. This would suggest that *MG828507* lncRNA may be a pivotal placental factor in the pathophysiology of pre-eclampsia.

There are some limitations in this study. First, the sample size is small in the association study between *MG828507* expression and the genetic variants. To demonstrate that these genetic variants act as an eQTL, further studies involving a large sample size are required. Second, in the functional experimental study, although overexpression of exogenous *MG828507* did not affect the *FLT1* expression level, it is still possible that induction of endogenous *MG828507* expression might increase the *FLT1* expression. Altogether, it is still noteworthy that the strong correlation of expression levels of *MG828507* and *FLT1* genes might implicate a presence of a common, underlying regulatory mechanism between the *MG828507* and *FLT1* genes.

In summary, genomic variants located between the *MG828507* and *FLT1* genes may affect their expression and, thus, exert influence on the severity of the fetal and placental symptoms in pre-eclampsia. Further characterization of this lncRNA gene is likely to elucidate the etiological role of the *FLT1* gene in pre-eclampsia within a genomic context.

## Figures and Tables

**Figure 1 jcm-11-04603-f001:**
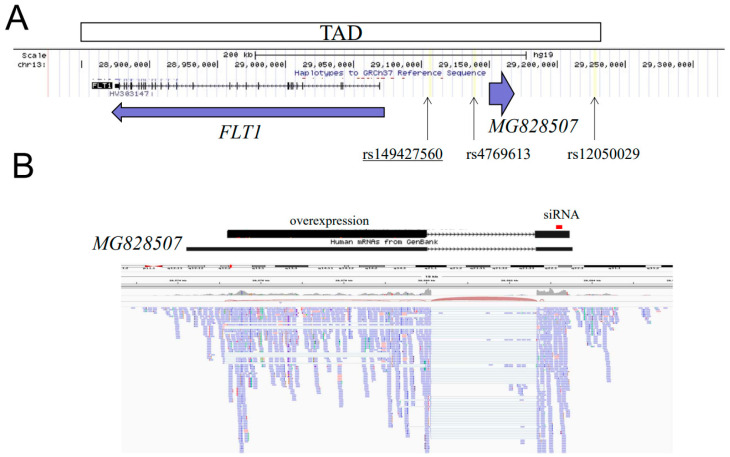
Genomic structure of the *FLT1* upstream region. (**A**) Location of the *MG828507* and *FLT1* genes. The horizontal arrows indicate their transcription direction. The extent of the TAD that incorporates these genes is indicated by the large box. The vertical arrows indicate the location of the three variants tested in this study. The STR, s149427560, is underlined. (**B**) RNA-seq of *MG828507*. The horizontal bar indicates the *MG828507* gene registered in the public database with its exon-intron structure. The upper panel indicates the position of the cDNA for overexpression experiments as well as the position targeted by the siRNA oligonucleotide used in this study.

**Figure 2 jcm-11-04603-f002:**
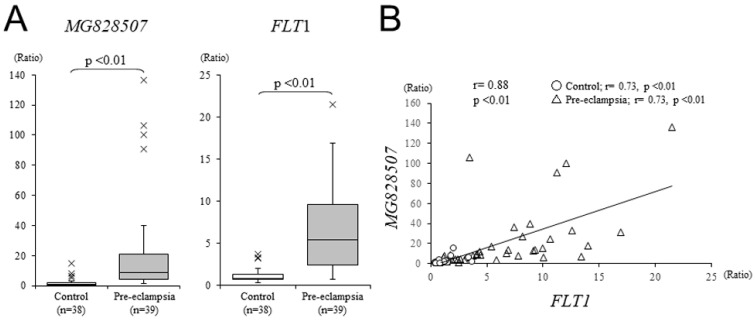
*MG828507* and *FLT1* expression in pre-eclamptic and normotensive placentas. (**A**) Differential expression of *MG828507* and *FLT1* in uncomplicated pregnancies (left) and in the pre-eclampsia samples (right). The boxes indicate the 25th and 75th percentiles, whilst the bands near the middle indicate the median values. The bars indicate the 1.5 interquartile ranges, with the outliers specifically marked. (**B**) Analysis of *MG828507* and *FLT1* correlations. Open circles indicate the uncomplicated pregnancy controls, and open triangles indicate pre-eclampsia. Regression lines are shown with correlation coefficients and *p* values.

**Figure 3 jcm-11-04603-f003:**
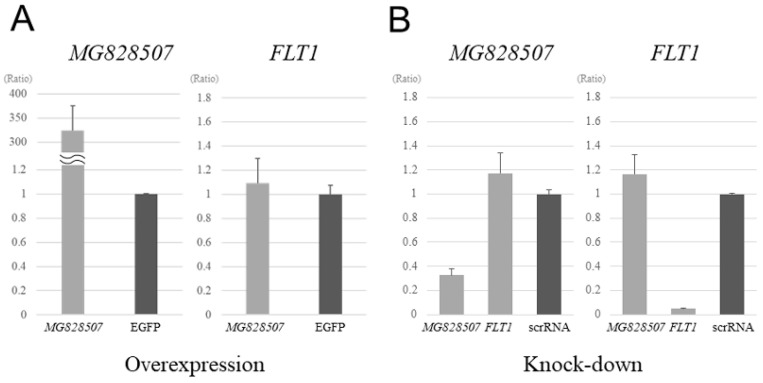
Analysis of the overexpression and knockdown of *MG828507*. (**A**) Expression of *MG828507* and *FLT1* after the exogenous overexpression of *MG828507*. HTR-8 trophoblast cells were transfected with an *MG828507* expression vector. The *MG828507* and *FLT1* transcript levels were analyzed by qRT-PCR. The vertical axis indicates the ratio relative to the data of cells transfected with control vector. (**B**) Expression of *MG828507* and *FLT1* after the knockdown of *MG828507*. HTR-8 trophoblast cells were transfected with an siRNA targeting *MG828507*. The *MG828507* and *FLT1* transcript levels were again analyzed by qRT-PCR. The siRNA for *FLT1* as well as a scrambled siRNA control were also used independently as control transfection experiments. The vertical axis indicates the ratio relative to the data of cells transfected with control oligonucleotides.

**Figure 4 jcm-11-04603-f004:**
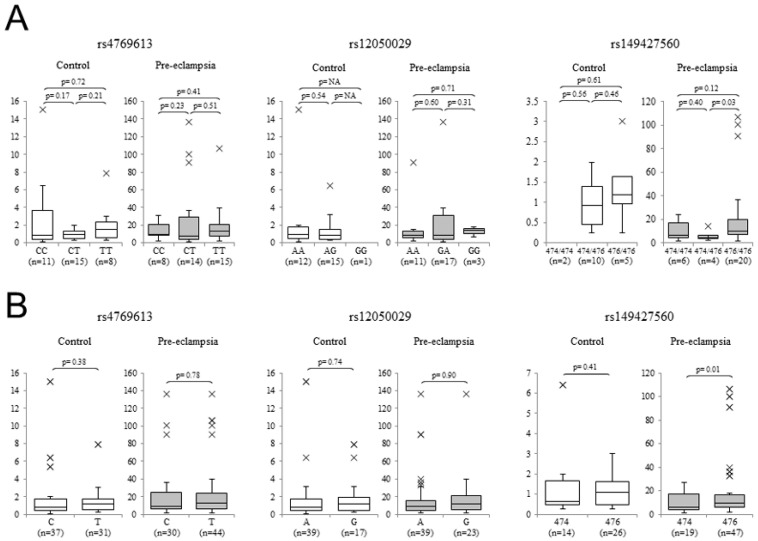
Correlations between *MG828507* expression and the indicated genetic variants. (**A**) Comparison of the placental mRNA levels by variant genotype. Data for rs4769613 (left), rs12050029 (center), and rs149427560 (right) are shown. In each panel, data for the normotensive placental controls are shown on the left, and those from pre-eclamptic cases are shown on the right. (**B**) Comparison of the placental mRNA levels by variant allele type. The boxes indicate the 25th and 75th percentiles. The bands near the middle indicate the median values. The bars indicate the 1.5 interquartile ranges, with the outliers specifically marked. Sample numbers and *p* values are shown in each panel.

**Figure 5 jcm-11-04603-f005:**
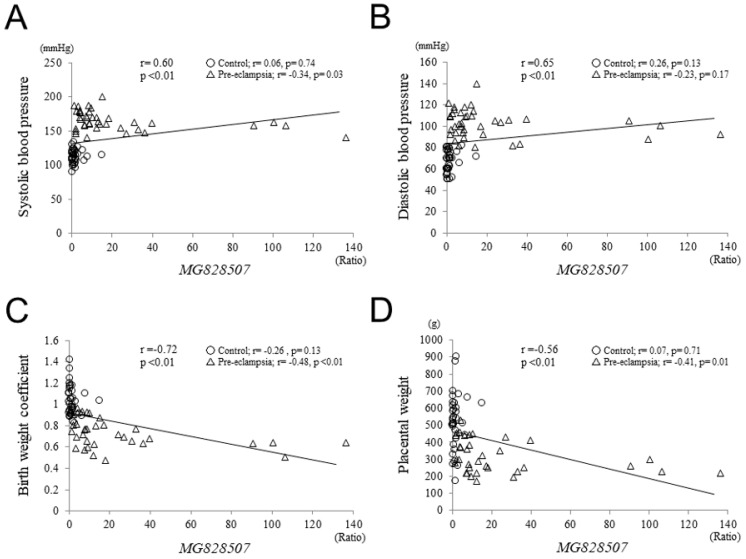
Correlations between *MG828507* expression and systolic blood pressure (**A**), diastolic blood pressure (**B**), normalized birth weight (**C**), or placental weight (**D**). Open circles indicate control uncomplicated pregnancies, and open triangles indicate pre-eclampsia. Regression lines are shown with correlation coefficients and *p* values.

**Table 1 jcm-11-04603-t001:** Characteristics of the normotensive and pre-eclamptic study subjects.

	Normotensive Pregnancy	Pre-Eclampsia	*p* Value
	*n* = 38	*n* = 39	
Maternal age (y)	30 (28–36) ^†^	30 (28–32)	n.s
Gestational age (weeks)	37 (31–38)	33 (32–36)	n.s
Systolic BP (mmHg)	114 (107–118)	163 (157–175)	<0.05
Diastolic BP (mmHg)	70 (60–76)	103 (93–110)	<0.05
Proteinuria ^‡^	0 (0%)	39 (100%)	<0.05
Body mass index (BMI) ^§^	21.2 (19.7–22.8)	21.0 (19.0–23.9)	n.s
Birth weight (g)	2649 (1827–3105)	1494 (1146–1974)	<0.05
Birth weight coefficient	1.000 (0.919–1.112)	0.717 (0.640–0.813)	<0.05
Placental weight (g)	540 (485–630)	300 (240–398)	<0.05

^†^ Data are the median (interquartile range). ^‡^ ≥2 g in a 24-h collection. ^§^ pre-pregnancy.

## Data Availability

The data presented in this study are available on request from the corresponding author.

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
