# Peer review of "Characterization of the MG828507 lncRNA Located Upstream of the FLT1 Gene as an Etiology for Pre-Eclampsia"

_jcm, 2022, doi:10.3390/jcm11154603_

Round 1

Reviewer 1 Report

The regulation of functional noncoding RNA can alter gene activity, modulating gene expression, transcription, and protein translation. This manuscript explores the role of a lncRNA MG828507 in preeclampsia. The importance of the topic cannot be overstated since preeclampsia is one of the most important causes of maternal deaths around the globe. In addition, lncRNAs may serve as potential diagnostic biomarkers in the future. 

The authors hypothesize that increasing the expression of the FLT1 gene likely predisposes the individual to pre-eclampsia.  They performed genome-wide expression profiling by RNA-seq of pre-eclamptic and normotensive placentas. In addition, the authors evaluated whether the MG828507 lncRNA regulates the expression of the FLT1 gene through in vitro experiments. 

The article is well written, treats an actual problem, and provides evidence about different expressions of lncRNA regulating SFLT1 in preeclampsiaThe authors found that MG828507 is abundant in the placenta, and the word is scarcely expressed in other tissues. The MG828507 lncRNA was highly expressed in the preeclamptic placenta compared to an uncomplicated normotensive pregnancy—notably, the MG828507 mRNA level correlated with that of the FLT1 in the placentas of pre-eclamptic women. 

Although the authors did not find a relation between the expression and the lncRNA, I believe that selecting patients with no clear criteria for severe preeclampsia might influence these negative results. I encouraged the authors to enhance the patients' clinical phenotype to assess the condition's severity better. For instance, it would be essential to classify whether they were early-onset versus late-onset preeclampsia. 

The authors should identify the most critical steps in the introduction and methodology necessary for the reader to understand the process and for peers to replicate their findings. The rest of the information could be allocated in the supplemental material.

Minor suggestions:

1.   In the abstract, please, modify the directly associated by “is involved” since, although a large amount of evidence has associated sFLT-1 with preeclampsia, a causal relationship has not been established.

2.   In the abstract, please modify the subject of study. The authors performed an RNA-seq of placentas from mothers who developed preeclampsia and controls. 

3.   The definition of preeclampsia has evolved, and I respectfully suggest the authors update the Is outdated by 20 years). 

4.   I believe that the authors should expand to the general reader the implications of the technology employed and the potential advantages over previous evaluations of gene and RNA expression in the placenta. An introduction about noncoding RNA and long noncoding RNA will be general to understand the manuscript's relevance.

5.   It is essential to describe the indications for delivery in normotensive patients in patients without labor. These controls are indeed complicated to obtain, and they should not be termed uncomplicated since they were delivered preterm for a specific reason.

6.   According to my previous commentary, the clinical criteria to define severe preeclampsia are outdated since proteinuria was removed from the criteria for severe preeclampsia a long time ago. 

7.   It is interesting that all the data is expressed in mean and standard deviation since biological data is frequently not normalized; I wonder whether the authors evaluated the normality of the data before comparing the groups.

8.   I am missing a section comparing to other authors exploring the role of other lncRNA in preeclampsia.

9.   Please add a section on strengths and limitations.

Author Response

To Reviewer#1

Major suggestions:

Thank you for valuable comments. Basically, we agree with the reviewer’s comment that the criteria for patient selection is not clear. We collected samples from women with severe PIH from 2005 to 2014 according to the criteria used in Japan at that time (Takagi et al. 2015). Although we know the diagnostic criteria changed in 2018, we used the old criteria in this study cohort. We excluded other type of PIH such as gestational hypertension, superimposed pre-eclampsia, eclampsia and mild PIH from the study. We have added study period and some comments regarding to the used criteria to the Materials and methods (lines 81 and 85-86).

As the reviewer suggested, we have added the analyses between MG828507 levels and other clinical parameters that indicate the disease severity including early or late onset, platelet count, transaminase or serum creatinine levels. However, there was no correlation between these parameters and MG828507 levels. We have added a comment in the Results (lines 284-287) and showed the data as a Supplementary Figures.

As the reviewer suggested, we have added the statement regarding the general information of lncRNAs as well as RNA-seq technology in the Introduction (lines 67-73). The most important step to identify the MG828507 gene was written in the Results section, not in the Materials and Methods (lines 188-199).

Minor suggestions:

  1. According to the reviewer’s suggestion, we have changed the description (line 16).
  2. According to the reviewer’s suggestion, we have changed the description (lines 17-18).
  3. We collected samples from women with severe PIH from 2005 to 2014 according to the criteria used in Japan at that time (Takagi et al. 2015). Although we know the diagnostic criteria changed in 2018, we used the old criteria in this study cohort. We excluded other type of PIH such as gestational hypertension, superimposed pre-eclampsia, eclampsia and mild PIH from the study. We have added study period and some comments regarding to the used criteria to the Materials and methods (lines 81 and 85-86).
  4. We agree with the reviewer’s comments. As the reviewer suggested, we have added the statement regarding the general information of lncRNAs as well as RNA-seq technology in the Introduction (lines 67-73).
  5. Normotensive subjects underwent a Caesarean section due to a breech presentation or a previous Caesarean section. In addition, we collected preterm normotensive control samples from pregnancies with a premature rupture of the membrane and underwent a Caesarean section due to a breech presentation or a previous Caesarean section without evidence of intrauterine infection. We have added the statement in Materials and Methods (lines 88-92).
  6. We collected samples from women with severe PIH from 2005 to 2014 according to the criteria used in Japan at that time (Takagi et al. 2015). Although we know the diagnostic criteria changed in 2018, we used the old criteria in this study cohort. We excluded other type of PIH such as gestational hypertension, superimposed pre-eclampsia, eclampsia and mild PIH from the study. We have added study period and some comments regarding to the used criteria to the Materials and methods (lines 81 and 85-86). In addition, recent study re-evaluated the importance of proteinuria as an indicator of severity classification of pre-eclampsia (Okamoto et al. Pregnancy Hypertens 2022).
  7. As the reviewer suggested, we evaluated the normality of the clinical parameters in Table 1, but some parameters did not show normal distribution. We have revised the data presentation in the Table 1 using median with the interquartile range.
  8. According to the reviewer’s suggestion, we have added the data of other lncRNA studies in the Discussion (lines 294-296).
  9. According to the reviewer’s suggestion, we have added a paragraph describing the strengths and limitations of this study in the Discussion (lines 354-362).

Reviewer 2 Report

The authors report data on a RNAseq analysis done in placentas from preeclamptic and control cases and found a highly significant overexpression of MG828507 in the preeclamptic cases. However, the authors could not demonstrate a correlation to clinical features of preeclampsia except placental and fetal size. The paper is straight forward written and clear in presenting the rsults. However, the importance of the findings are not convincingly presented. Also a discussion of other RNAseq approaches on preeclamptic placentas is missing. Nevertheless the paper is as it is consistent and of good quality. Data of course can not be changed or improved. 

Author Response

To Reviewer#2

Thank you for valuable comments. Although the importance of the MG828507 gene should be presented more, the data cannot be changed, as the reviewer stated. Instead, we have added a paragraph describing the strengths and limitations of this study in the Discussion to enhance the importance of the MG828507 gene (lines 354-362). In addition, we have added the data of other recent RNA-seq studies for pre-eclampsia in the Introduction according to the reviewer’s suggestion (lines 70-73).